# Performance Robustness of AI Planners
## to Changes in Software Environment

**Chris Fawcett**
University of British Columbia
fawcettc@cs.ubc.ca

**Mauro Vallati**
University of Huddersfield
m.vallati@hud.ac.uk

**Alfonso E. Gerevini**
Università degli Studi di Brescia
alfonso.gerevini@unibs.it

**Holger H. Hoos**
University of Leiden
hh@liacs.nl

### Abstract

Solver competitions have been used in many areas of AI to assess the current state of the art, and to guide future research and real-world applications. AI planning is no exception, and the International Planning Competition (IPC) has been frequently run for nearly two decades. Due to the organizational and computational burden involved with running these competitions, solvers are generally compared using a single homogeneous software environment for all competitors.

In this work, we use the competing planners and benchmark instance sets from the 2014 IPC Agile and Optimal tracks to investigate two questions. First, how is planner performance affected by the specific choice of software environment? Second, is it a good strategy to run planners with more recent versions of their software dependencies, in order to maximise performance? By running these competition tracks on eight distinct software environments, we show that planner performance varies significantly based on the chosen software environment, that the magnitude of this variation differs considerably between planners, and that using more recent software versions is not always beneficial.

## Introduction

Automated Planning has been studied extensively for several decades, resulting in automated planners being deployed in a variety of real-world applications. Part of the considerable progress in developing powerful domain-independent planners can be ascribed to the International Planning Competitions (IPCs). Competitions play an important role in many areas of AI, helping to drive forward research and development of solvers for prominent problems, as well as incentivising the development and distribution of related tools and benchmarks. Competitions also play a prominent role in assessing and improving the state of the art in solving challenging AI problems, such as planning.

While all of the planning systems participating in the IPC are available to be used after the competition, top-ranked planners receive much of the attention and often have considerable long-term impact on research as well as on real-world applications. For this to make sense, we need to assume that, at least from a qualitative point of view, conclusions derived from competition results generalise well to other – even significantly different – hardware and software environments than those used for running the competition. It is well-known that competition results are already strongly affected by the set of benchmark instances and the evaluation function used to assess planner performance, as well as by the way in which benchmarks are described, and by the set of competitors (Howe and Dahlman 2002; Long and Fox 2003; Hoffmann and Edelkamp 2005; Gerevini et al. 2009; Linares López, Celorrio, and Olaya 2015a; Vallati and Vaquero 2015). It also comes as no surprise that results are strongly affected by resource bounds – in particular, the running time cutoff and the amount of RAM available to the planner. Moreover, an analysis performed on the SAT competition showed that ranks of solvers are also affected by pseudo-random number seeds used in runs of randomised solvers (Hurley and O'Sullivan 2015).

Interestingly, a previous investigation performed by Howe and Dahlman (2002) showed that the relative (qualitative) performance of planners can vary, among other factors, also when run using different hardware environment configurations. However, as their work focused on identifying potential sources of performance variation, their analysis concentrated on assessing differences between two different machines having the same software environment configuration.

In this work, we present an investigation into the impact of software environment choices on competition outcomes. Our experimental analysis involves eight different software configurations, including the choice of C/C++ compiler version, Python interpreter version and Java version. We attempt to identify aspects that have unequal impact on planner performance, in order to emphasise those aspects that have to be carefully considered when interpreting competition results.

When dealing with such software configuration choices, a question naturally arises:

*Is it better to use the latest available versions of software environment components such as compilers or interpreters, or stick to the specific environment used during the IPC?*

To help address questions such as this, we investigate the performance variation of planners that took part in two deterministic tracks of the 2014 International Planning Competition: the *Optimal* and *Agile* tracks. The *Optimal* track is one of the longest-standing tracks in the IPC series, with many participating planners and substantial impact on the field of AI planning. While the *Agile* track was new for IPC 2014, its emphasis on planner running time and low resource requirements made it ideally suited for our analysis.

Our results show that competition rankings are strongly

affected by the software environment. For a more extensive analysis, which shows also the impact of different hardware configurations on planners, the interested reader is referred to (Bocchese et al. 2018).

## Sources of Performance Variation

In this work, given the lack of analysis in the literature, we focus on the impact of software environment on planners' performance. However, there are many possible sources of performance variation that can affect empirical performance analyses and competition outcomes. Here, we briefly survey some of the most important of these sources of variation.

**Planner randomization**. Many planners take advantage of randomization to improve average-case performance and to avoid manual deterministic development choices. This randomization can result in very different planner trajectories in repeated runs with different random seeds, with a correspondingly wide variation in the resulting performance (Hurley and O'Sullivan 2015).

**Running time and memory**. Generally, increasing the running time or memory allocated to a planning system will result in more problem instances solved. However, this performance improvement will not be uniform across planners; for example, planners that perform extensive precomputation or use pattern databases tend to benefit more from increased memory limits (Linares López, Celorrio, and Olaya 2015b).

**Hardware environment**. It is clear that hardware choices, such as CPU type and speed, can affect planner performance, and it is known that planners are affected to varying degree by such differences in hardware environment (Howe and Dahlman 2002). Other aspects of the hardware environment that can have significant impact on planner performance include CPU cache, memory bandwidth, local storage medium, general-purpose graphics processing units (GPGPU) and network fabric.

**Software environment**. There are many aspects of the software environment configuration that can affect planner performance. These choices include the operating system and version, versions and compilation of system libraries (e.g., LIBC), as well as the version, linking, and compiler used for building all further software components required by a given planning system.

**Benchmark instances**. The instances used for evaluating planning systems should be challenging and need to allow for performance differences between planners to be identified. While it is clear that the use of different benchmarks can lead to very different results, we note that planning instances are often created using randomised generators, where a few parameters define the size and the difficulty of the resulting instances. The choice of problem instance domains, generator settings, as well as instance set size and distribution will all have an effect on planner performance (Howe and Dahlman 2002). Furthermore, also the order in which elements are listed in benchmarks has strong influence on planner performance (Vallati et al. 2015b; Vallati and Serina 2018).

**Ranking mechanism**. The metric used to assess planner performance (running time, instance set coverage, solution quality) and the techniques for aggregating performance

|     | GCC   | Python | JVM |
|-----|-------|--------|-----|
| gpj | 4.7.2 | 2.7.3  | 1.7 |
| Gpj | 4.8.2 | 2.7.3  | 1.7 |
| gPj | 4.7.2 | 2.7.10 | 1.7 |
| GPj | 4.8.2 | 2.7.10 | 1.7 |
| gpJ | 4.7.2 | 2.7.3  | 1.8 |
| GpJ | 4.8.2 | 2.7.3  | 1.8 |
| gPJ | 4.7.2 | 2.7.10 | 1.8 |
| GPJ | 4.8.2 | 2.7.10 | 1.8 |

Table 1: The 8 software configurations considered in this investigation. Lowercase and uppercase are used to distinguish the "base" and "newer" configurations of each component.

across a given set of benchmark instances affect the outcome of empirical performance evaluations and competitions. Some competitions use an absolute scoring mechanism (such as mean running time), while others (such as earlier editions of the IPC) use relative scoring mechanisms, where the performance score of a planner is potentially affected by the performance of its competitors. The performance of a given planning system in relation to others can vary considerably depending on the ranking mechanism used (Linares López, Celorrio, and Olaya 2015b; Vallati, Chrpa, and McCluskey 2018).

## Methodology

For our experimental analysis of the impact of the software environment on planner performance, we chose two sequential, deterministic tracks of the 2014 International Planning Competition (IPC): the *Agile* (15 participants) and *Optimal* (17 participants) tracks. These two tracks provide a very interesting test-bed, as they rank competitors using nearly opposite metrics. The competing planners are therefore likely to exploit significantly different approaches and techniques. In the *Optimal* track, planner running time is of limited importance: planners are assessed according to their ability to generate optimal solution plans within a given (large) cutoff time. In the *Agile* track, on the other hand, the quality of solutions is irrelevant, as planners are ranked according to their ability to quickly find a solution.

We chose to investigate three major software components in our analysis: GCC compiler version, Python interpreter version, and Java version. Nearly every planner which took part in IPC 2014 was entirely or partially reliant on components compiled with GCC, and different compiler versions are very likely to produce different executables even when identical command-line options are used. We selected GCC versions 4.7.2 and 4.8.2 as the two configuration options, since 4.7.2 was that used in the competition and several of the planners did not successfully compile with GCC versions more recent than 4.8.2.

Python and Java were by far the next most common software dependencies for the planners we considered. We selected Python 2.7.3 and Oracle Java 1.7.0_45, the versions used in IPC-2014, as well as Python 2.7.10 and Oracle Java 1.8.0_65, the most recent versions at the time of our experiments under which planners would run successfully.

The combination of these choices resulted in 8 poten-

tial software configurations, all of which were used in this work. Table 1 details each of the 8 configurations considered. For conciseness, each configuration will be referred to using one letter for each component: g for the GCC version, p for the Python version, and j for the Java virtual machine version. Since we consider two versions of each component, we use lowercase for indicating the *base* version, and uppercase when referring to the more recent version of the given component. Therefore, we denote the default configuration provided by the organisers of IPC 2014 as *gpj* (the *base* configuration), and the configuration with the more recent of each option as *GPJ* (the *newer* configuration).

Many of the planners that took part in the tracks we considered did require some modification in order to run successfully on our hardware and software configurations, for example to avoid writing temporary files into their source directories and polluting results when executing runs concurrently. We consider these modifications minor and do not believe that they had any effect on planner execution or running times. There were two exceptions, namely the *Freelunch* planner from the *Agile* track and the *AllPaca* planner from the *Optimal* track. In the case of *Freelunch*, we could not successfully run the planner on our computer cluster with any version of Java. As far as we can determine, this was caused by the high-memory shared environment on each cluster node, as *Freelunch* would crash immediately on launch with a Java JVM memory allocation exception. In the case of *AllPaca*, the planner relied on the presence of a specific commercial Lisp variant, and we were unable to modify it to work with any of the Lisp distributions available on our systems. These two planners have therefore been removed from our results, but we fully expect that if these issues were to be fixed, they would not significantly impact our results.

All the experiments were run on a cluster of homogeneous machines running CentOS version 5.0, each containing two Intel Xeon X5650 2.66GHz six-core processors with 12MB cache, and 24GB of available RAM. All planner runs were performed independently in parallel, with each run assigned one CPU core, 8GB of RAM, and the running time limits used in each track of IPC 2014. Running time and memory limits were monitored and enforced using tools from AClib (Hutter et al. 2014).

It is common practice in automated planning to include randomised components in planning systems. Randomisation is useful, e.g., for breaking ties during the heuristic search, introducing some noise in the heuristic search state evaluation, or performing search restarts. Evidently, planner performance can be affected by this source of stochasticity. In order to account for this and attempt to isolate the impact of software configuration on planner performance, our results for any given planner were obtained by averaging over five independent runs on each benchmark instance.

## Empirical Analysis

Table 2 illustrates how the instance set coverage of the *Optimal* track planners are affected by our software configurations. Several planners exhibit a sizeable performance drop when Java 1.8 is used instead of version 1.7. The planners most affected by this are *MIPlan* and *NuCeLaR*; the plan-

| Instance coverage for each software configuration | | | | | | | | |
|---|---|---|---|---|---|---|---|---|
| Planner | gpj | Gpj | gPj | GPj | gpJ | GpJ | gPJ | GPJ |
| cGamer-bd | 130.2 | **131.4** | **131.4** | 130.8 | 128.2 | 127.4 | 127 | 128.8 |
| RIDA | 117.2 | 115.8 | **117.4** | 116 | **117.4** | 116.2 | 117 | 116.4 |
| Metis | 112.2 | 112.4 | 112.6 | 112.6 | **112.8** | 112.6 | 112.4 | 111.8 |
| SymBA-1 | 108.2 | 108.6 | 108.4 | 108.4 | **108.8** | 108.6 | 108.4 | 108.6 |
| SymBA-2 | 108 | 108.8 | 108.2 | 108.6 | 108.6 | **109** | 108.4 | 108.4 |
| MIPlan | 112.6 | 112.6 | **113.2** | 113 | 102.2 | 103.2 | 103.4 | 103 |
| D-Gamer | 107.6 | **110** | 109 | 106.6 | 104.2 | 105.6 | 104.6 | 105.4 |
| NuCeLaR | **108.8** | 108.6 | 108.2 | 108.6 | 98.6 | 98.6 | 98.4 | 98.8 |
| DPMPlan | 101.4 | 101.8 | 102.2 | 102.2 | 102.4 | **102.8** | 102.4 | 102.4 |
| Cedalion | 95.4 | 96 | 96 | **96.2** | 95.2 | 95.8 | 95.6 | 95.8 |
| Gamer | **96.8** | 96.4 | 96.4 | 96.4 | 93.2 | 92.6 | 91.8 | 93.8 |
| Rlazya | 91 | 91.8 | 92.2 | 91.8 | 91.8 | 92 | **92.6** | 91.8 |
| SPMaS | 75.8 | 76.8 | 74.6 | 75.4 | **77.2** | 75.6 | 74.8 | 75.4 |
| Hflow | 56 | 57 | 56.8 | **57.2** | 56 | **57.2** | 56.6 | 57 |
| Hpp | 15 | 15 | 15 | 15 | 15 | 15 | 15 | 15 |
| Hpp-ce | 15 | 15 | 15 | 15 | 15 | 15 | 15 | 15 |

Table 2: Number of problem instances solved (instance coverage) for each of our 8 software configurations, using the IPC 2014 *Optimal* track planners and benchmark instance set. We present the mean coverage over 5 independent runs. Boldface is used to indicate the best performance achieved by a planner, in presence of variations.

| IPC score for each software configuration | | | | | | | | |
|---|---|---|---|---|---|---|---|---|
| Planner | gpj | Gpj | gPj | GPj | gpJ | GpJ | gPJ | GPJ |
| Cedalion | 107.4 | 106.2 | 106.9 | 107.6 | 105.8 | 107.1 | **107.6** | 107.0 |
| IBaCoP | **70.1** | 65.8 | 70.0 | 64.9 | 69.1 | 66.0 | 69.8 | 66.2 |
| USE | 79.3 | 77.7 | **79.6** | 78.1 | 78.1 | 78.1 | 77.6 | 75.7 |
| ArvandHerd | 89.2 | **90.2** | 88.1 | 88.2 | 87.9 | 87.5 | 88.8 | 89.8 |
| IBaCoP2 | 59.6 | 56.6 | **60.0** | 55.7 | 58.1 | 55.6 | 58.6 | 56.1 |
| Jasper | **84.8** | 83.7 | 84.3 | 79.9 | 82.3 | 82.3 | 84.5 | 79.7 |
| Mercury | **67.8** | 67.5 | 67.8 | 66.5 | 66.4 | 67.7 | 67.1 | 67.0 |
| BFS-f | 66.1 | 66.4 | 66.2 | **66.7** | 64.7 | 66.1 | 66.1 | 66.5 |
| Probe | 71.4 | 71.2 | 71.3 | 71.4 | 70.7 | 71.4 | **71.5** | 71.2 |
| YAHSP3 | 73.7 | 73.6 | 73.5 | 73.2 | 73.3 | 73.1 | **73.8** | 73.5 |
| Madagascar-pc | **63.4** | 62.6 | **63.4** | 62.6 | 63.1 | 62.6 | 63.2 | 62.4 |
| YAHSP3-mt | **56.0** | 54.2 | 54.2 | 53.8 | 53.7 | 54.4 | **56.0** | 53.4 |
| Madagascar | **64.1** | 63.9 | 64.0 | 63.9 | 63.9 | **64.1** | 64.0 | 63.3 |
| SIW | 49.6 | 50.2 | 50.1 | 49.8 | 49.4 | **50.3** | 50.0 | 50.1 |

Table 3: IPC score for each of our 8 software configurations, using the IPC 2014 *Agile* track planners and benchmark instance set. We present the mean coverage over 5 independent runs. Where performance variations exist for a given planner, we show the best performance achieved by that planner in boldface.

ners based on *Gamer*, i.e., *Gamer*, *cGamer-bd* and *Dynamic-Gamer* also show a performance drop, although not as significant as for the previously-mentioned planners. We note that the performance variations observed for *MIPlan*, *NuCeLaR*, *Dynamic-Gamer* and *Gamer* were sufficient to cause changes in competition ranking between software environment configurations. The remaining planners of the *Optimal* track show minor performance fluctuation. Figure 1(a) provides a visual representation of the performance variation on each of the 8 considered configurations.

Table 3 shows how the software configuration affects planner performance for the *Agile* track. Performance is measured in terms of IPC score [1], which is focused on the running time required to find any satisficing plan; the quality of those plans is not considered. Some planners from this track demonstrated high sensitivity to the GCC com-

---

[1]https://helios.hud.ac.uk/scommv/IPC-14/rules.html

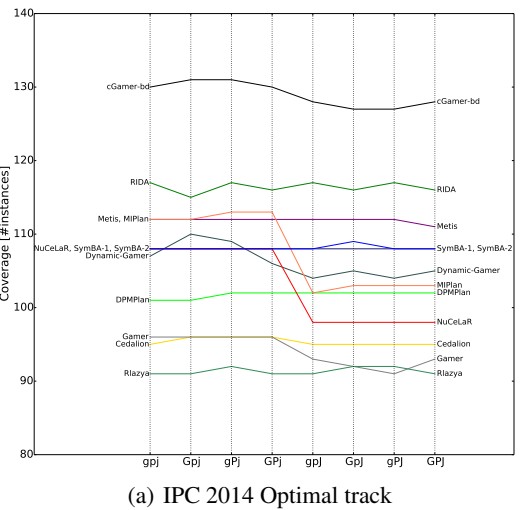

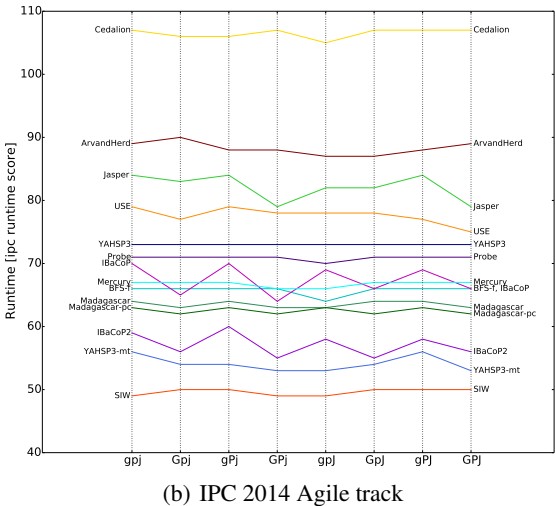

(a) IPC 2014 Optimal track      (b) IPC 2014 Agile track

Figure 1: Mean number of instances solved (instance coverage) and IPC score, respectively, using the planners and benchmark instance sets from the IPC 2014 *Optimal* 1(a) and *Agile* 1(b) tracks. We present results for each of the 8 software configurations.

piler version. For example, extreme variation can be observed in the performance of *IBaCoP* and *IBaCoP2*, to the extent of causing changes in competition ranking for *IBaCoP*. The performance of other *Agile* track planners, in particular *Use* and *Jasper*, are affected by a combination of GCC and Python versions. The other planners exhibit only minor performance variation.

Figures 1(a) and 1(b) show how the competition ranks are affected by the eight considered software configurations. It is easy to observe that the ranks are significantly influenced and many of the planners face changes in rank. Moreover, these results provide a valuable example of the impact of other sources of variation on the reproducibility of competition results: almost all of the considered planners were ranked differently in IPC 2014 official results.

Intuitively, one would expect the competing planners to achieve their best performance either on the *base* configuration (*gpj*) or on the *newer* configuration (*GPJ*). With regard to the former, the underlying hypothesis is that the planners' code should have been somehow "tuned" – for the sake of competition performance – for the configuration used in IPC 2014; this appears plausible, since participants had access to the competition cluster for testing their planners (Vallati et al. 2015a). On the other hand, it is also likely that optimisations introduced in newer versions of compilers or interpreters will be reflected in noticeable performance improvements of planning systems making use of them. Interestingly, for planners that participated in the *Agile* track, we observed a tendency for the *base* configuration to lead to the best performance (but still for only 6 of 14 competitors), while none of the planners achieved their best performance using the *newer* configuration. The situation is different for the *Optimal* track planners; Table 2 indicates that, while no planner achieves its best performance on the *newer* configuration, the *base* configuration is rarely the one providing the best results. Each planner is affected in a different way by

the considered 8 software configurations.

## Conclusion

Our analysis shows that the software environment in which a planner is run can substantially impact performance, and that this effect varies significantly among planners. These software environment changes can be as minor as the version of the compiler used to create each planner executable. Our analysis also indicates that running planners on software configurations that are more recent than those used in the competition can have surprisingly detrimental effects on performance.

While our experimental observations do suggest that competition performance results should be carefully interpreted, we caution that these observations should not be misunderstood as invalidating or diminishing the utility of planner competitions in general. Attempting to compensate for many of the sources of performance variation discussed in this paper would place a heavy burden on competition organisers, both in terms of time and additional computational resources. Allowing competitors the ability to customize their own software configuration for the competition, as done in the 2018 IPC using containers, could potentially reduce this source of variation, but could also have the side effect of biasing the competition results in favour of competitors with the expert knowledge, computational resources and time to finely tune their systems.

We see several possible avenues for future work: first, using the knowledge gained in this work, the study and development of a competition measuring solver performance across several distinct hardware and software environments; second, a thorough analysis of additional sources of performance variation not covered in (Bocchese et al. 2018), including benchmark instance set selection and solver stochasticity.

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
