# OpenReview forum: "Performance Robustness of AI Planners to Changes in Software Environment"
_icaps-conference.org/ICAPS/2019/Workshop/WIPC_

### Official Review · AnonReviewer3 · 2019-04-17
**Interesting paper but hard to interpret results**

**Rating:** 7
**Confidence:** 4

**Review:**

The paper points out that the software environment, in particular the
versions of compilers and interpreters, can have an effect on planner
performance. The analysis is done on the optimal and agile tracks of
the IPC 2014.

It would be interesting to see an updated version of the numbers for
the 2018 IPC. An important change in 2018 was the use of software
containers that (1) would allow easy control over the software
environment for an experiment like the one done in this paper, and (2)
allows the participants to control their own software environment. The
authors discuss the latter as a hypothetical option but since it was
actually used in IPC 2018, the text could be updated accordingly. One
other important change in 2018 was that the satisficing and agile
tracks did not use a relative scoring method. The paper mentions that
"the IPC" used relative scoring but is not true for 2018 (and older
IPCs as well).

The actual evaluation is somewhat hard to interpret for several reasons:
1) the tables and plots cannot be easily "filtered" for one dimension.
    I would appreciate a single plot/table for Java, one for Python,
    and one for C++ where the effect of that choice can be seen.
2) It is not clear which changes are significant and which ones are
    not. There obviously are some other factors influencing the results
    besides the software versions, so the numbers vary naturally. For
    example, there are differences of around 1 point that change with
    the Java version in planners that do not use Java, so obviously a
    difference of 1 point cannot be significant. Have you done a
    statistical analysis of these results and could you add details
    about this to the paper?
3) The investigated software versions were released one (C++), two
    (Java), and three (Python) years apart. For the C++ the changes
    between the versions are relatively minor. Even for planners that
    do not use explicit randomization, changing minor things in the
    code can change the way the binary is aligned in memory, which can
    affect pointer values and in turn the ordering of iterating over hash
    maps (for example). This can easily lead to performance differences
    as the one you observed. My point is that I am not convinced that
    the differences in the compiler version have an actual effect or if
    the differences are caused by other low level changes. In the
    example above, the same effect could be reached for example by
    adding a global variable to the code. Changes like this are not
    something planners could or should reasonably be tuned for.
4) Finally, and perhaps most importantly, the results differ
    drastically from the ones reported in the IPC. You mention this in
    the paper but only as a side remark about reproducibility. The
    difference between the results here and the IPC results is about 50
    tasks (SymBA*'s coverage goes down by 43 tasks, cGamer's coverage
    goes up by 10). The largest difference you report in the paper is a
    factor of 4 away from this. So there seems to be something that
    influences the performance of the planners drastically. Without
    further information on what that is, it is hard to judge if the
    differences you report could not also be caused by this unknown
    factor.

A small side remark on freelunch: it had the same issue in IPC 2018.
Java doesn't play well in a memory-limited environment
(https://superuser.com/questions/472587/why-does-limiting-my-virtual-memory-to-512mb-with-ulimit-v-crash-the-jvm).
The solution there was to set the internal memory limit of freelunch
significantly lower than the overall limit (using "-Xmx5000m" on a 8 GB
limit).

Despite the negative points above, I think this paper would spark an
interesting discussion at the workshop so it should be accepted. I
would appreciate comments on the issues above, though.

---

### Official Review · AnonReviewer2 · 2019-04-25
**Two points to clarify**

**Rating:** 7
**Confidence:** 4

**Review:**

This paper looks at how the performance of IPC2014 planners depends on the software toolchain installed on the machine -- in particular, the version of GCC, Python and the JVM used.

Two things that it would be useful to clarify in the paper, and which I hope are not too onerous a burden:

1) Do all the presented planners use GCC /and/ Python /and/ Java?

Please clarify either way. Otherwise, if there's no difference between e.g. gpj and gpJ, we don't know if it's due to the planner not using Java.

2) Do you have any insights into what causes the performance differences?

My intuition is that changing the toolchain may have effects on memory layout, and hence tie-breaking or other matters usually considered to be implementation details.  Not all the planners will necessarily offer up statistics to support this, but for instance, is there a planner that reports the number of nodes generated/expanded/evaluated (planners based on fast downward do this, for instance), and does this ever change when different versions of gcc/python/Java are used?  Or is it mainly that e.g. 'this version of gcc optimizes the code of planner X more or less well than the other version'?